# Effect of Sleep Quality on Anxiety and Depression Symptoms among College Students in China’s Xizang Region: The Mediating Effect of Cognitive Emotion Regulation

**DOI:** 10.3390/bs13100861

**Published:** 2023-10-20

**Authors:** Yingting Wang, Zixuan Guang, Jinjing Zhang, Lixin Han, Rongqiang Zhang, Yichun Chen, Qi Chen, Zhenjia Liu, Yuan Gao, Ruipeng Wu, Shaokang Wang

**Affiliations:** 1Key Laboratory for Molecular Genetic Mechanisms and Intervention Research on High Altitude Disease of Tibet Autonomous Region, School of Medicine, Xizang Minzu University, Xianyang 712082, China; 2Key Laboratory of High Altitude Hypoxia Environment and Life Health, School of Medicine, Xizang Minzu University, Xianyang 712082, China; 3School of Public Health, Xi’an Jiaotong University Health Science Center, No.76 Yanta West Road, Xi’an 710049, China; 4Disease Control and Prevention Division, Shaanxi Provincial Health Commission, No.112 Lianhu Road, Xi’an 710003, China; 5School of Public Health, Shaanxi University of Chinese Medicine, Xianyang 712046, China; 6Key Laboratory of Environmental Medicine and Engineering of Ministry of Education, Department of Nutrition and Food Hygiene, School of Public Health, Southeast University, Nanjing 210009, China

**Keywords:** sleep quality, cognitive emotion regulation, anxiety symptoms, depressive symptoms, Chinese college students

## Abstract

Background: While the exact mechanisms are not fully understood, there are significant links between sleep quality, anxiety, depressive symptoms, and cognitive emotion regulation. This research examines how sleep quality affects anxiety and depressive symptoms, as well as the potential of cognitive emotion regulation strategies (CERS) to moderate the impact of sleep quality on these symptoms. Methods: The Chinese version of the Pittsburgh Sleep Quality Index (CPSQI), the Cognitive Emotion Regulation Questionnaire (CERQ), the Patient Health Questionnaire-9 (PHQ-9), and the Generalized Anxiety Disorder Scale-7 (GAD-7) were all completed online by students from two colleges in China’s Xizang region. Results: The study included 4325 subjects. The prevalence of poor sleep quality, anxiety symptoms, and depression symptoms was 45.69%, 36.81%, and 51.86%, respectively. We observed significant direct effects on poor sleep and severity of anxiety/depression: c’1 = 0.586 (0. 544–0.628), and c’2 = 0.728 (0.683–0.773). Adaptive CERS only had a mediating effect on the relationship between sleep quality and depression symptoms, with a1b3 = −0.005 (−0.011–−0.001). The link between poor sleep quality and the intensity of anxiety and depression was significantly affected by the indirect effects of maladaptive CERS: effect a2b2 = 0.126 (0.106–0.147), and effect a2b4 = 0.145 (0.123–0.167). Conclusions: Individuals who experience poor sleep quality are more likely to have increased levels of anxiety and depression. However, enhancing sleep quality led to a decrease in anxiety and depression levels. Adaptive CERS did not predict anxiety, but they did predict depression. Multiple maladaptive CERS could increase levels of anxiety and depression. To prevent mental stress, it is crucial to examine sleep problems among college students, understand their cognitive strategies, promote the adoption of adaptive CERS, and reduce the reliance on maladaptive CERS.

## 1. Introduction

The most prevalent mental illnesses among college students are anxiety and depression, with prevalence rates of depressive and anxiety symptoms of 33.6% and 39.0%, respectively [1]. These disorders seriously affect the learning ability, academic performance, interpersonal relationships, and future professional development of college students [2]. The relationship between sleep disorders and depression and anxiety is bidirectional, with each playing a crucial role in the occurrence and maintenance of the other [3]. Studies have reported that 90% of patients with severe depression have sleep disorders [4], and sleep disorders are more common in adolescents with depressive symptoms [5]. Xu et al. also found that sleep disturbance is closely related to depressive symptoms, that a lack of sleep may lead to depressive symptoms [6], and that sleep disorders are major predictors of future depression [7]. The prevalence of anxiety among insomnia patients is 45–70% [8,9]. Anxiety can have a negative impact on sleep quality, leading to a decrease in overall quality of life and an increased risk of developing anxiety [10,11].

Children who exhibit emotional and behavioral issues tend to have greater difficulties with sleep [12]. Additionally, biological data indicate a strong connection between sleep and mood [13]. At present, there are nine recognized cognitive emotional regulation strategies (CERS), among which five help to reduce unpleasant states caused by stressful experiences (i.e., adaptive CERS): acceptance, putting into perspective, positive refocusing, refocusing on planning, and positive reappraisal [14]. Adaptive CERS help people face trauma rather than escape it. The other four methods aggravate unpleasant states caused by stressful experiences (i.e., maladaptive CERS): rumination, catastrophizing, self-blame, and blaming others [14]. Adaptive CERS can reduce susceptibility to stress insomnia, but maladaptive CERS can increase it [15].

Depression is believed to be characterized by impaired cognitive emotion regulation [16]. The subscores of self-blame, rumination, catastrophizing, and positive reappraisal were found to predict depressive and anxiety symptoms in healthy college students of various races [17]. Kraaij and Garnefski conducted a multidisciplinary study of chronic diseases and found that CERS are crucial in depression [18]. The employment of a catastrophizing method by diabetic patients predicts the existence of depressed symptoms, while positive reappraisal may benefit diabetic patients [19]. Maladaptive CERS and anxiety and sleep quality are positively correlated, and adaptive CERS can directly or indirectly affect sleep quality through factors other than anxiety [20]. A study of healthy young women found that the use of adaptive CERS is negatively correlated with depressive symptoms, and depression may partially mediate the link between maladaptive CERS and poor sleep quality [21]. Some researchers applied group counseling based on cognitive emotion regulation theory as an intervention for college students and found that depression was effectively relieved, the scores of positive refocusing increased, and the scores of ruminating and blaming others decreased. This suggests that strengthening some cognitive coping mechanisms may help with depressive mood [22].

While the exact mechanisms are not fully understood, there are significant links between sleep quality, anxiety, depressive symptoms, and CERS. Most prior research that examined the connection between CERS and sleep quality did so in pairs rather than by including it in a comprehensive model. For this reason, the current study aimed to investigate how sleep quality affects symptoms of anxiety and depression and the mediating effect of CERS in college students in China’s Xizang region; for the hypotheses, see Figure 1.

## 2. Materials and Methods

### 2.1. Participants

Student participants in this study came from two universities in Xizang, China. This cross-sectional survey was administered on Questionnaire Star, an online questionnaire survey platform in China, from June 2021 to July 2021. The sampling method used was cluster convenience sampling, and the link to the questionnaire was given to the instructor, who clearly explained the research purpose to students. Afterward, the subjects voluntarily and anonymously completed the questionnaire. Only a scientific investigation was conducted using the data that were acquired, which were held in complete confidence. There were 4885 questionnaires gathered in all, of which 560 questionnaires, which either had short response times or the same choices selected for each item, were considered invalid. With an overall recovery rate of 88.5%, 4325 valid questionnaires were collected.

### 2.2. Sleep Quality

The Chinese version of the Pittsburgh Sleep Quality Index (CPSQI) [23], developed by Buysse et al. [24] and refined by Tsai et al. [23], was used to assess the individuals’ sleep quality over the previous month. The 19 questions in the scale are divided into seven categories, including daytime dysfunction, latency to fall asleep, sleep length, sleep quality, sleep disorders, hypnotic medications, and sleep efficiency. When evaluating sleep quality, the individual scores for each component (which fall between 0 and 3) are amalgamated. The final sleep score can range from 0 to 21, with elevated scores indicating poorer sleep quality. PSQI > 5 is defined as having poor sleep quality [23,24].

### 2.3. Cognitive Emotion Regulation

We employed the Chinese version of the Cognitive Emotion Regulation Questionnaire (CERQ-C) [25] to assess the participants’ usage of CERS after a negative occurrence. The CERQ is a 36-item questionnaire with nine conceptually separate subscales, among which 5 help to reduce unpleasant states caused by stressful experiences (i.e., adaptive CERS): acceptance, putting into perspective, positive refocusing, refocusing on planning, and positive reappraisal [14]. Another 4 methods aggravate unpleasant states caused by stressful experiences (i.e., maladaptive CERS): rumination, catastrophizing, self-blame, and blaming others [14]. The CERQ assessment has nine subscales, each comprising four items. Each item is graded on a Likert scale of 1 (rarely) to 5 (usually always), with 5 being the highest score. The adaptive CERS score ranges from 20 to 100, while the maladaptive CERS score ranges from 16 to 80. To obtain the total CERQ score, the scores of all nine subscales are add up. The total score ranges from 36 to 180, and a higher subscale score indicates that the individual uses the strategy more frequently.

### 2.4. Symptoms of Anxiety

To determine anxiety symptoms, the Generalized Anxiety Disorder Scale-7 (GAD-7) was implemented [26]. This scale has been proven reliable in the Chinese population [27]. It consists of seven items that are rated on a four-point Likert scale, with scores ranging from 0 to 3. Respondents are asked to assess how often they experienced anxiety symptoms over the last two weeks. The score of the scale should be between 0 and 21, with a score from 0 to 5 indicating no anxiety symptoms, a score from 6 to 9 indicating mild anxiety, a score from 10 to 14 indicating moderate anxiety, and a score from 15 to 21 indicating severe anxiety symptoms [28]. The higher the score, the more severe the anxiety symptoms.

### 2.5. Depression Symptoms

To evaluate depressive symptoms, the Patient Health Questionnaire-9 (PHQ-9) employs a 4-point Likert scale with nine items (0 = none, 1 = some days, 2 = more than half of the days, and 3 = almost every day). The PHQ-9’s overall score ranges from 0 to 27 based on the severity of depressive symptoms experienced by the respondent over the previous two weeks: 0–4 is normal, 5–9 is mild depressive symptoms, 10–14 is moderate depressive symptoms, 15–19 is moderate severe depressive symptoms, and 20–27 is severe depressive symptoms [29].

### 2.6. Other Variables

We also gathered information on age, residential location (1 = urban, 2 = rural), only child status (1 = yes, 2 = no), sex (1 = male, 2 = female), relationship quality (1 = good, 2 = normal, and 3 = poor), academic pressure (1 = mild, 2 = moderate, and 3 = severe), cigarette usage (1 = yes, 2 = no), ethnicity (1 = Han Chinese, 2 = Tibetan, and 3 = other), and alcohol consumption (1 = yes, 2 = no).

### 2.7. Statistical Analysis

The basic demographic parameters, CPSQI scores, CERS, and degrees of anxiety and depression were all described using descriptive statistics. Qualitative data were described using the mean (SD) and rate or constituent ratio. To explore the connections among variables, we utilized Pearson’s correlation analysis. All of the above statistical analyses were conducted using SPSS 26.0 for Windows, developed by IBM Corp., in Armonk, NY, USA. In addition, we constructed and assessed a parallel mediation model using Model 4 of the SPSS macro PROCESS version 3.3, which was developed by Preacher and Hayes. The number of bootstrapping samples was 5000, and bootstrap 95% confidence intervals that did not include 0 were taken to indicate statistical significance. The CPSQI score (sleep quality) was included as the independent variable, adaptive and maladaptive CERS were set as the mediating variables, and anxiety and depression were set as the dependent variables. All models controlled for age, sex, ethnicity, residential location, only child status, family relationship quality, academic pressure, smoking status, drinking status, and BMI. The difference was statistically significant (*p* < 0.05).

## 3. Results

### 3.1. Demographic Characteristics

The participants’ demographic characteristics are presented in Table 1. The study included 4325 subjects. The participants’ mean (SD) age was 19.90 (1.34) years, and there were 1668 male participants (38.60%). Among the participants, 1743 (40.30%) were Han Chinese, 2470 (57.10%) were Tibetan, 1210 (28.00%) were urban residents, and 3887 (89.90%) had harmonious family relations. A total of 2006 (46.40%) students reported severe academic pressure, 1716 (39.70%) students reported moderate academic pressure, 890 (20.60%) students reported smoking, and 2308 (53.40%) students reported drinking. The average BMI was 21.37 (3.44). The average CPSQI score was 5.54 (2.78). The prevalence of poor sleep quality, anxiety symptoms, and depression symptoms were 45.69%, 36.81%, and 51.86%, respectively. The average adaptive and maladaptive CERS scores were 63.49 (10.11) and 42.43 (8.43), respectively.

### 3.2. Correlation Analysis

There were positive correlations of sleep quality with adaptive CERS (r = 0.103, *p* < 0.01), maladaptive CERS (r = 0.321, *p* < 0.05), anxiety (r = 0.497, *p* < 0.05), and depression (r = 0.537, *p* < 0.05). Adaptive CERS scores were positively correlated with maladaptive CERS (r = 0.534, *p* < 0.05), anxiety (r = 0.213, *p* < 0.05), and depression (r = 0.179, *p* < 0.05). Maladaptive CERS scores were positively correlated with anxiety (r = 0.434, *p* < 0.05) and depression (r = 0.433, *p* < 0.05). There was also a positive correlation between anxiety and depression (r = 0.806, *p* < 0.05).

The bivariate correlations between CERQ subscale scores and sleep quality, anxiety symptoms, and depression symptoms are displayed in Table 2. The bivariate correlation between CERQ subscale scores and sleep quality were as follows (from strongest to weakest): catastrophizing (r = 0.288, *p* < 0.05), rumination (r = 0.242, *p* < 0.05), blaming others (r = 0.214, *p* < 0.05), putting into perspective (r = 0.207, *p* < 0.05), self-blame (0.199, *p* < 0.05), acceptance (r = 0.099, *p* < 0.05), positive refocusing (r = 0.075, *p* < 0.05), refocus on planning (r = 0.040, *p* < 0.05), and positive reappraisal (r = −0.035, *p* < 0.05). The bivariate correlations between the CERQ subscales scores and symptoms of anxiety were as follows (from strongest to weakest): catastrophizing (r = 0.366, *p* < 0.05), rumination (r = 0.354, *p* < 0.05), self-blame (r = 0.296, *p* < 0.05), putting into perspective (r = 0.285, *p* < 0.05), blaming others (r = 0.260, *p* < 0.05), positive refocusing (r = 0.183, *p* < 0.05), acceptance (r = 0.164, *p* < 0.05), refocus on planning (r = 0.119, *p* < 0.05), and positive reappraisal (0.033, *p* < 0.05). The bivariate correlations between CERQ subscales scores and symptoms of depression were as follows (from strongest to weakest): catastrophizing (r = 0.376, *p* < 0.05), rumination (r = 0.341, *p* < 0.05), putting into perspective (r = 0.297, *p* < 0.05), self-blame (r = 0.285, *p* < 0.05), blaming others (r = 0.269, *p* < 0.05), positive refocusing (r = 0.162, *p* < 0.05), acceptance (r = 0.148, *p* < 0.05), refocus on planning (r = 0.069, *p* < 0.05), and positive reappraisal (−0.009, *p* > 0.05).

### 3.3. Results of the CERS Mediating Effect Analysis

The parallel mediation analysis is shown in Figure 2. Sleep quality significantly positively predicted adaptive CERS (β = 0.266, 95% CI = 0.156–0.376), maladaptive CERS (β = 0.849, 95% CI = 0.759–0.935), anxiety (β = 0.712, 95% CI = 0.670–0.753), and depression (β = 0.867, 95% CI = 0.822–0.912). Adaptive CERS did not predict anxiety (β = −0.0004, 95% CI = −0.013–0.013) but did predict depression (β = −0.019, 95% CI = −0.033–−0.005). Maladaptive CERS positively predicted anxiety (β = 0.148, 95% CI = 0.132–0.164) and depression (β = 0.171, 95% CI = 0.153–0.188).

The mediating effect of adaptive strategies was limited to the relationship between poor sleep quality and depression (a1b3: effect = −0.005, 95% CI = −0.011–−0.001). There were significant indirect effects of maladaptive CERS on the relationship between poor sleep quality and anxiety/depression) (a2b2: effect = 0.126, 95% CI = 0.106–0.147, and a2b4: effect = 0.145, 95% CI = 0.123–0.167). The direct effects of poor sleep quality on anxiety/depression severity were significant (c’1 = 0.586, 95% CI = 0.544–0.628, and c’2 = 0.728, 95% CI = 0.683–0.773, respectively) (see Table 3).

### 3.4. Specific Types of CERS

Poor sleep quality had a significant indirect effect on anxiety severity through CERS, such as self-blame, acceptance, and rumination. Except for refocusing on planning, CERS had a significant partial mediating influence on the association between poor sleep quality and anxiety symptoms. The association between depression and sleep quality was significantly impacted indirectly by CERS, such as self-blame, acceptance, and rumination. CERS had a significant partial mediating influence on the association between poor sleep quality and depression symptoms, except for refocusing on planning and positive reappraisal. (see Table 4).

## 4. Discussion

The results of the correlation analysis showed that sleep quality, adaptive CERS, maladaptive CERS, anxiety, and depression were positively correlated. The parallel mediation analysis showed that maladaptive CERS partially mediated the relationships between poor sleep quality and anxiety/depression. Adaptive CERS partially mediated the association between poor sleep quality and depression, whereas the relationship between poor sleep quality and anxiety was not significantly affected by adaptive CERS. Poor sleep quality may also be associated with higher levels of anxiety and depression, and CERS may play a partial mediating role in this relationship. The use of adaptive CERS did not affect anxiety but reduced the level of depression.

Poor sleep habits, such as sleep deprivation and daytime sleepiness, were associated not only with lower ratings of their current quality of life and educational environment but also with symptoms of anxiety and depression [30]. Sleep quality is also closely related to students’ academic performance [31]. Students with poor sleep quality and poor sleep habits may be unable to concentrate on their studies, which may not only affect their physical health in the long run but also create a vicious cycle of poor academic performance, depression/anxiety, and poor sleep quality. Many studies have shown that insufficient sleep and poor sleep quality are closely related to mental health problems. Ineffective sleep patterns and poor sleep quality can worsen depression and anxiety symptoms, and they increase the likelihood of developing such symptoms [32,33]. Psychosocial factors are closely related to depressive symptoms, but the influencing factors differ in different countries. Attachment and religious belief are more closely related to depressive symptoms in Western countries, while emotional and social support are related to depressive symptoms in Eastern countries, such as in Indonesia with the elderly [34]. Depressive symptoms were also associated with short stature, overweight, age, gender, and other physiological indicators and population characteristics [35]. Depressive symptoms may have a stronger association with life satisfaction in adolescents [36]. At present, the widely accepted pathogenesis of depression includes monoaminergic theory, neuron/synaptic remodeling theory, and immune and inflammatory theory [37]. Studies have shown that depression in patients is accompanied by circadian rhythm disruption, daily functional disturbance, and metabolic disturbance, and these circadian rhythm disturbances usually return to normal after a depressive episode [38]. Drugs that regulate circadian rhythm disorders have become strategies for the treatment of depression, and melatonin is one of them. It plays a crucial role in regulating circadian rhythm. Major depressive disorder is a syndrome of insufficient secretion of melatonin, so insufficient secretion of melatonin is used as a biomarker of depression [39]. Melatonin drugs simultaneously affect three pathogenetic theories of depression [37]. Other studies have shown that neuromodulators such as serotonin are important factors in stress-induced mood disorders such as depression. Serotonin is the basic substance of sleep and depression, sleep deprivation is also a major symptom of depression, and sleep quality can predict depression [40]. Anxiety or depressive symptoms can predict poor sleep quality in adolescents [41]. Sleep quality is closely related to social behaviors, and poor sleep quality may lead to reduced personal interaction and social withdrawal [32].

Sleep disorders are related to impaired emotional regulation and coping strategies, and lack of sleep plays an important role in emotional regulation [42]. The adaptive regulation of emotion, which enables individuals to better control and manage emotional experiences or current conditions, is the key to daily adaptation to environmental factors [43]. Emotional dysregulation may lead individuals to experience feelings of loneliness, peer conflict, or feelings of hopelessness, which are all associated with anxiety and depression [44,45]. In the development and maintenance of emotional disorders, including anxiety and depression, CERS play a cross-diagnostic function. While adaptive CERS lessen anxiety, maladaptive CERS increase stress related to COVID-19 [46]. While most people tend to utilize the same form of CERS in response to life events, studies have indicated that depressed and anxious patients demonstrate more rumination and less reappraisal [47]. In addition, adaptive CERS do not have long-lasting effects [48]. For people with difficulties in cognitive emotion regulation, especially after experiencing a stressful event, there is a bias toward negative thoughts, as well as difficulties in information processing and cognitive evaluation, generating a vicious cycle of long-term repeated negative thoughts [49]. Maladaptive CERS, such as rumination and catastrophizing, affect mental health, rendering individuals prone to depression and anxiety [50]. In contrast, adaptive CERS can protect against mental illness in adverse situations [51]. A systematic review of the relationship between emotion regulation strategies and depressive and anxiety symptoms in adolescents has shown a correlation. Maladaptive CERS scores are positively connected with psychological distress (anxiety/depression), with rumination demonstrating the largest positive association. Adaptive CERS scores are inversely correlated with both of these symptoms, with acceptance exhibiting the strongest negative correlation [52]. Additional research has revealed correlations between self-blame, positive reappraisal, and positive refocusing and depressive symptoms. The use of catastrophizing and blaming others was associated with anxiety symptoms [53]. Self-blame, acceptance, ruminating, catastrophizing, and blaming others were significantly positively correlated with depression and anxiety in Japanese people, according to a meta-analysis of Japanese samples, while positive refocusing, refocusing on planning, positive reappraisal, and putting things into perspective were significantly negatively correlated with depression and anxiety [54]. Similar results were obtained in the current Chinese sample. More frequent use of strategies such as self-blame, rumination, catastrophizing, and acceptance and less frequent use of positive refocusing was positively correlated with the onset and severity of depression; additionally, more frequent use of strategies such as rumination, catastrophizing, and blaming others, and less frequent use of positive refocusing were associated with severe anxiety symptoms. Moreover, researchers believe that acceptance should be regarded as a maladaptive CERS [55]. According to Kraaij et al., positive refocusing and positive reappraisal were found to be negatively correlated with depressive symptoms in adolescents with type 1 diabetes, while acceptance, rumination, catastrophizing, self-blame, and blaming others were positively correlated with depressive symptoms [56]. The correlations between maladaptive CERS and anxiety and depression in the present study were consistent with previous findings, while adaptive CERS were also positively correlated with anxiety and depression, although the correlations were weak. CERS usage may be influenced by sociodemographic factors, including age, sex, and educational attainment, with participants with different sociodemographic characteristics adopting different adjustment strategies in the face of life events. For example, older people have more life experience, so they may use more effective adaptive CERS. Women are more likely to employ acceptance, rumination, positive refocusing, and putting into perspective, while men are more likely to use blaming others. Highly educated people are more likely to ruminate, refocus on planning and engage in positive reappraisal. Subjects with less education scored higher on self-blame, catastrophizing, and blaming others [46,57,58].

A simple mediation analysis examined the mediating effects of nine CERS on the relationships between poor sleep quality and psychological distress (anxiety/depression); poor sleep quality had a significant indirect effect on anxiety severity through self-blame, acceptance, rumination, and other CERS. Except for refocusing on planning, CERS had a significant partial mediating influence on the association between poor sleep quality and anxiety symptoms. CERS, such as self-blame, acceptance, and rumination, had significant indirect effects on the relationship between sleep quality and depression severity. CERS had a significant partial mediating influence on the association between poor sleep quality and depression symptoms, except for refocusing on planning and positive reappraisal. This may be related to the fact that adaptive CERS, such as reappraisal, acceptance, and problem-solving, can protect people from mental illness in the face of adverse situations [51] and that two types of adaptive CERS, positive reappraisal and refocusing, do not have long-lasting effects [48]. Studies have found that using positive reappraisal as an emotion regulation strategy alleviates or even eliminates pain and stress. This may be because positive reappraisal can effectively relieve negative emotions and allow individuals to reevaluate the situation [59]. There is complexity and flexibility in emotion regulation. Each individual may be inclined to adopt multiple strategies to regulate emotion in the face of negative events, and there may be complex interactions among various strategies [60].

There are a few limitations to this study. First, the ability to establish causal relationships among variables was limited by the cross-sectional design of this study. Future longitudinal research will be required to evaluate the causal connections between the various factors. Second, every piece of data was self-reported, and there may be a reporting bias in participants whose anxiety or depressive symptoms affect their CERS. Third, sleep quality was only investigated in the form of a questionnaire, some students may not pay much attention to sleep-related information at ordinary times, and the content filled in may be different from the actual situation. To ensure the accuracy of information, it is better to use objective recording tools such as electronic bracelets to ensure the authenticity of the information. Fourth, the concept of emotion regulation is broad, but this study only focused on CERS. Last, this study only included college students, which may limit the generalizability of the findings. However, with a large sample size, this study looked at the overall impact of sleep quality and CERS on anxiety and depressive symptoms.

## 5. Conclusions

In conclusion, the present study examined the direct effects of sleep quality on the symptoms of anxiety and depression and the indirect effects of CERS on college students in China’s Xizang region. Individuals who experience poor sleep quality are more likely to have increased levels of anxiety and depression. Adaptive CERS did not predict anxiety, but it did predict depression. The use of adaptive CERS did not affect anxiety but decreased the level of depression. Maladaptive CERS had a positive predictive effect on psychological distress (anxiety/depression), indicating that the use of multiple maladaptive CERS could increase levels of anxiety and depression. Therefore, it is necessary to explore sleep problems in college students, understand individual cognitive strategies, help individuals adopt adaptive CERS, and reduce the use of maladaptive CERS to prevent psychological distress.

## Figures and Tables

**Figure 1 behavsci-13-00861-f001:**
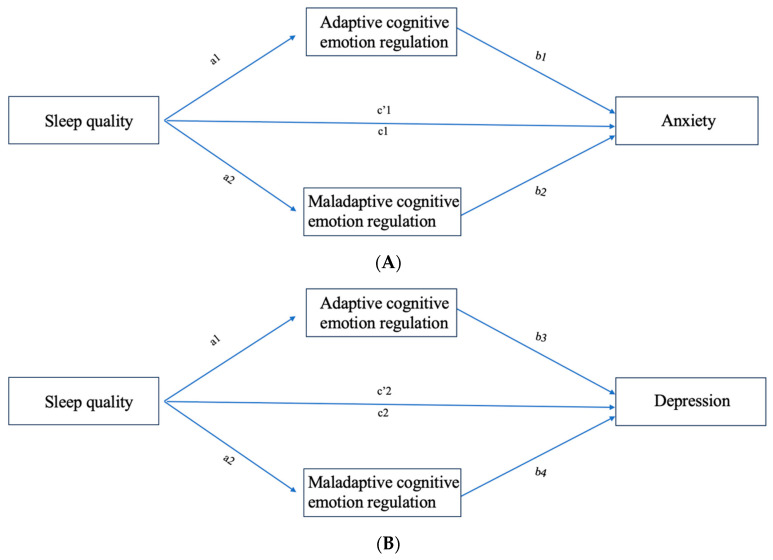
(**A**,**B**) The research model.

**Figure 2 behavsci-13-00861-f002:**
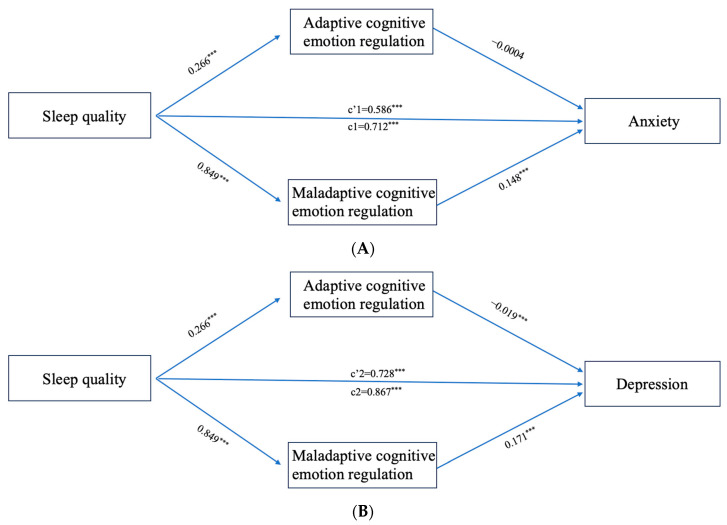
(**A**,**B**) Mediating effect of CERS on the relationship between sleep quality and psychological distress (anxiety (**A**) and depression (**B**)). Path coefficients are shown in unstandardized regression coefficients: c = total effect, and c’ = direct effect. The covariates were age, residential location, only child status, sex, relationship quality, academic pressure, smoking, ethnicity, drinking, and BMI. *** *p* < 0.001.

**Table 1 behavsci-13-00861-t001:** General characteristics of the participants.

Variable	Category	Number of Samples	Mean (SD)/Percentage (%)
Age		4325	19.90 (1.34)
Gender	Male	1668	38.60
	Female	2657	61.40
Ethnicity	Han Chinese	1743	40.30
	Tibetan	2470	57.10
	Other	112	2.60
Residential location	Urban	1210	28.00
	Rural	3115	72.00
Only child status	Yes	866	20.00
	No	3459	80.00
Family relations	Good	3887	89.90
	General	340	7.90
	Poor	98	2.30
Academic pressure	Mild	603	13.90
	Moderate	1716	39.70
	Severe	2006	46.40
Smoking status	No	3435	79.40
	Yes	890	20.60
Alcohol consumption	No	2017	46.60
	Yes	2308	53.40
BMI		4325	21.37 (3.44)
CPSQI score		4325	5.54 (2.78)
	Normal	2349	54.31
	Poor sleep quality	1976	45.69
GAD-7 score	Normal	2733	63.19
	Mild	1111	25.69
	Moderate	344	7.95
	Severe	137	3.17
PHQ-9 score	normal	2082	48.14
	Mild	1479	34.20
	Moderate	549	12.69
	Moderate severe	160	3.70
	Severe	55	1.27
CERQ-M		4325	42.43 (8.43)
CERQ-A		4325	63.49 (10.11)

**Table 2 behavsci-13-00861-t002:** Correlation among sleep quality, cognitive emotion regulation strategies, depression, and anxiety.

Variable	1	2	3	4	5	6	7	8	9	10	11	12	13
1. CPSQI	1												
2. SB	0.199 **	1											
3. Acc	0.099 **	0.522 **	1										
4. Rum	0.242 **	0.491 **	0.449 **	1									
5. P-Ref	0.075 **	0.364 **	0.433 **	0.528 **	1								
6. R-Plan	0.040 **	0.353 **	0.410 **	0.372 **	0.506 **	1							
7. P-Reap	−0.035 *	0.274 **	0.387 **	0.292 **	0.478 **	0.757 **	1						
8. PP	0.207 **	0.311 **	0.271 **	0.409 **	0.287 **	0.154 **	0.129 **	1					
9. Cat	0.288 **	0.272 **	0.171 **	0.419 **	0.208 **	0.054 **	−0.027	0.607 **	1				
10. BO	0.214 **	0.243 **	0.175 **	0.327 **	0.182 **	0.045 **	0.001	0.514 **	0.598 **	1			
11. CERQ-A	0.103 **	0.509 **	0.701 **	0.568 **	0.748 **	0.813 **	0.792 **	0.497 **	0.269 **	0.245 **	1		
12. CERQ-M	0.321 **	0.663 **	0.439 **	0.759 **	0.432 **	0.274 **	0.178 **	0.629 **	0.791 **	0.735 **	0.534 **	1	
13. GAD-7 score	0.497 **	0.296 **	0.164 **	0.354 **	0.183 **	0.119 **	0.033 *	0.285 **	0.366 **	0.260 **	0.213 **	0.434 **	1
14. PHQ-9 score	0.537 **	0.285 **	0.148 **	0.341 **	0.162 **	0.069 **	−0.009	0.297 **	0.376 **	0.269 **	0.179 **	0.433 **	0.806 **

CPSQI = Chinese version of the Pittsburgh Sleep Quality Index; SB = self-blame; Acc = acceptance; Rum = rumination; P-Ref = positive refocusing; R-Plan = refocus on planning; P-Reap = positive reappraisal; PP = putting into perspective; Cat = catastrophizing; BO = blaming others; CERQ-A = Adaptive Cognitive Emotion Regulation Questionnaire score; CERQ-M = Maladaptive Cognitive Emotion Regulation Questionnaire score; GAD-7 = Generalized Anxiety Disorder Scale-7; PHQ-9 = Patient Health Questionnaire-9. ** *p* < 0.001, * *p* < 0.05 level (2-tailed).

**Table 3 behavsci-13-00861-t003:** Total, direct, and indirect effects among the variables.

Path	Effect	Boot SE	95% CI	
			Lower	Upper
**Model ^a^**				
Total effect	0.712	0.021	0.670	0.753
Direct effect	0.586	0.021	0.544	0.628
Total indirect effect	0.126	0.010	0.107	0.145
Sleep quality→ CERQ_A→ anxiety	−0.00001	0.002	−0.004	0.004
Sleep quality→ CERQ_M→ anxiety	0.126	0.010	0.106	0.147
**Model ^b^**				
Total effect	0.867	0.023	0.822	0.912
Direct effect	0.728	0.023	0.683	0.773
Total indirect effect	0.139	0.010	0.119	0.160
Sleep quality → CERQ_A → depression	−0.005	0.002	−0.011	−0.001
Sleep quality → CERQ_M → depression	0.145	0.011	0.123	0.167

^a^ Results of the mediating effect of CERS on the relationship between sleep quality and anxiety; ^b^ Results of the mediating effect of CERS on the relationship between sleep quality and depression.

**Table 4 behavsci-13-00861-t004:** The results of the simple mediation analysis of nine CERS on the relationship between sleep quality and psychological distress (anxiety and depression).

Mediating Variable (M)	Effect of Sleep Quality on M (a)	Effect of M on Anxiety (b)	Direct Effect (c’)	Indirect Effect (ab)	Effect of M on Depression (b)	Direct Effect (c’)	Indirect Effect (ab)
SB	0.158 ***	0.306 ***	0.665 ***	0.048 ***	0.308 ***	0.820 ***	0.049 ***
Acc	0.071 ***	0.153 ***	0.703 ***	0.011 ***	0.138 ***	0.859 ***	0.010 ***
Rum	0.222 ***	0.347 ***	0.636 ***	0.077 ***	0.354 ***	0.790 ***	0.079 ***
P-Ref	0.053 ***	0.219 ***	0.702 ***	0.012 ***	0.208 ***	0.857 ***	0.011 ***
R-Plan	0.024 ***	0.119 ***	0.711 ***	0.003	0.055 **	0.867 ***	0.001
P-Reap	−0.045 **	0.060 ***	0.716 ***	−0.003 ***	0.006	0.869 ***	0.001
PP	0.166 ***	0.283 ***	0.666 ***	0.047 ***	0.326 ***	0.814 ***	0.054 ***
Cat	0.281 ***	0.321 ***	0.622 ***	0.090 ***	0.365 ***	0.766 ***	0.103 ***
BO	0.187 ***	0.238 ***	0.669 ***	0.045 ***	0.267 ***	0.818 ***	0.050 ***

SB = self-blame; Acc = acceptance; Rum = rumination; P-Ref = positive refocusing; R-Plan = refocus on planning; P-Reap = positive reappraisal; PP = putting into perspective; Cat = catastrophizing; BO = blaming others. *** *p* < 0.001, ** *p* < 0.01.

## Data Availability

The raw data supporting the conclusions of this article are available through Xizang Minzu University; contact Wu Ruipeng for access approval.

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
