# Peer review of "Effect of Sleep Quality on Anxiety and Depression Symptoms among College Students in China’s Xizang Region: The Mediating Effect of Cognitive Emotion Regulation"

_behavsci, 2023, doi:10.3390/bs13100861_

Round 1
Reviewer 1 Report
Thank you very much for this opportunity to revise the manuscript titled “Effect of sleep quality on anxiety and depression symptoms among college students in China’s Tibetan region: the mediating effect of cognitive emotion regulation” that was submitted to Behavioral Sciences.
I would like to appreciate you for performing the work on the important psychological problem. My comments which may help you to improve the manuscript are listed below:
1. Results of abstract should be more informative and very important findings should be mentioned, briefly.
2. It is unclear why the authors ignored confessional factors and indicators of physicality (e.g. height / body mass index), which in our opinion can have a significant impact on the prevalence of anxiety and depression symptoms, sleep disorders among population.
3. I would recommend to the authors to supplement the article with the following sources:
https://doi.org/10.1017/s0021932020000401
https://doi.org/10.1016/j.jad.2021.02.064
https://doi.org/10.3390/ijerph20116002
https://doi.org/10.3390/proceedings2022083019
4. Limitation of the framework and suggestions for future studies as well as practical implications should be added and expanded substantially.
5. List of abbreviations should be added as a supplementary file.
I am sure that the answers to these comments / questions will improve the quality of this article.
I will be glad to review the revised manuscript.
Reviewer 2 Report
Authors have done a cross sectional study on effect of sleep quality on anxiety and depressive symptoms in college students and looked into the mediating effects.
There hypothesis is sound and has a good rationale however methodology and outcomes are not. It should be clearly mentioned in the title and the manuscript that neither the effect of 2 variables or the good mediation analyses can be done in a cross sectional study. There can be correlations that can be deducted from this study which can suggest for future lingitudinal data collection. Correlation strenghts often do not show effects of one variable over other, neither can these analyses be used to suggest that one variable can be preditor of another as mentioned in the result section.
Reviewer 3 Report
The article reveals the topical issue of the interaction between anxiety and depression and sleep disturbances, which is of particular relevance to the student population. However, I have some comments that need to be corrected:
1) The authors report a significant number of invalid responses in the survey. It is worth adding an explanation of what invalid means. What criteria did the authors use to determine that a respondent's answer was invalid?
2) Why did the authors not use cut-offs for sleep quality, anxiety, and depression? It would be useful to demonstrate the frequency of poor sleep quality and different degrees of anxiety and depression.
3) Why do the authors use p<0.01 as a statistically significant value, while p<0.050 is generally accepted in medical research? Correlations should be presented for each indicator in the form (r=X, p=Y).
4) Table 1 is titled Demographic characteristics of participants, although it also includes clinical characteristics of respondents. The title should be changed to "General characteristics" or the data should be separated.
5) In Table 2, all variables should be listed with the abbreviation of the relevant scale, e.g. CPSQI sleep quality. All variables should be capitalised.
6) In the commentary to Table 4, it is inappropriate to highlight the ***<0.000 level. The commonly used designation is p<0.001. In all places, replace <0.05 level with p<0.05.
7) Line 256 - reference 30 is in the middle of the word.
8) The discussion should be expanded with a brief overview of potential mechanisms that may ensure the interaction between sleep and anxiety and depression, in particular, stress neuromodulators (DOI:10.1155/2020/1931737), the melatonergic system (DOI:10.1007/s11064-022-03646-5), etc. To emphasise the expediency of studying these relationships in general, it is worth noting that the effect of sleep on anxiety and depression in various diseases has been confirmed (DOI:10.5937/afmnai39-33652). It is also advisable to emphasise the need for this study in the student population because the parameters under study affect students' perception of the academic environment (DOI:10.1186/s12909-021-02544-8), which is one of the main factors of their academic performance and, consequently, the quality of the education system.
Reviewer 4 Report
I would like to congratulate the authors on an excellent cross-sectional study. The entire study design was well conducted, and pointing out all limitations that could cause bias in inferences was well appreciated. All analyzes were adequate for the objectives. The state of the art and the discussion were well conducted.
I would just like some information, if there was an adjustment for some variables, was the Bonferroni adjustment considered?
Furthermore, I congratulate on the excellent study. I am in favor of publication, and the study will contribute greatly to the scientific community and society.
Best regards!
Round 2
Reviewer 3 Report
The authors took into account all the necessary comments, which significantly improved the quality of the manuscript.